# Hepatoprotective Effect of Medicine Food Homology Flower Saffron against CCl_4_-Induced Liver Fibrosis in Mice via the Akt/HIF-1α/VEGF Signaling Pathway

**DOI:** 10.3390/molecules28217238

**Published:** 2023-10-24

**Authors:** Huajuan Jiang, Xulong Huang, Jiaxin Wang, Yongfeng Zhou, Chaoxiang Ren, Tao Zhou, Jin Pei

**Affiliations:** 1State Key Laboratory of Southwestern Chinese Medicine Resources, Chengdu 611137, China; jianghuajuan@stu.cdutcm.edu.cn (H.J.); huangxulong@stu.cdutcm.edu.cn (X.H.); chaoxiangren92@gmail.com (C.R.); 2Pharmacy College, Chengdu University of Traditional Chinese Medicine, Chengdu 611137, China; 3First Clinical Medical College, Zhejiang Chinese Medical University, Hangzhou 310053, China; 18868441510@163.com; 4The First Affiliated Hospital of Anhui University of Traditional Chinese Medicine, Hefei 230031, China; zhouyfyt@163.com

**Keywords:** saffron, nutritional value, hepatoprotective activity, AKT/HIF-1α/VEGF axis

## Abstract

Liver fibrosis refers to a complex inflammatory response caused by multiple factors, which is a known cause of liver cirrhosis and even liver cancer. As a valuable medicine food homology herb, saffron has been widely used in the world. Saffron is commonly used in liver-related diseases and has rich therapeutic and health benefits. The therapeutic effect is satisfactory, but its mechanism is still unclear. In order to clarify these problems, we planned to determine the pharmacological effects and mechanisms of saffron extract in preventing and treating liver fibrosis through network pharmacology analysis combined with in vivo validation experiments. Through UPLC-Q-Exactive-MS analysis, a total of fifty-six nutrients and active ingredients were identified, and nine of them were screened to predict their therapeutic targets for liver fibrosis. Then, network pharmacology analysis was applied to identify 321 targets for saffron extract to alleviate liver fibrosis. Functional and pathway enrichment analysis showed that the putative targets of saffron for the treatment of hepatic fibrosis are mainly involved in the calcium signaling pathway, the HIF-1 signaling pathway, endocrine resistance, the PI3K/Akt signaling pathway, lipid and atherosclerosis, and the cAMP signaling pathway. Based on the CCl_4_-induced liver fibrosis mice model, we experimentally confirmed that saffron extract can alleviate the severity and pathological changes during the progression of liver fibrosis. RT-PCR and Western blotting analysis confirmed that saffron treatment can prevent the CCl_4_-induced upregulation of HIF-1α, VEGFA, AKT, and PI3K, suggesting that saffron may regulate AKT/HIF-1α/VEGF and alleviate liver fibrosis.

## 1. Introduction

Liver fibrosis refers to a complex fibrotic inflammatory reaction caused by a variety of reasons, including viral infection, high-fat and high-sugar diet, alcohol, drugs, parasites, etc. [1]. It is usually characterized by an imbalance in the oxidative stress response, abnormal activation of hepatic stellate cells (HSCs), excessive deposition of extracellular matrix (ECM), and other abnormal liver tissue structures and behaviors [2]. Among them, about 40% of liver fibrosis diseases can evolve into end-stage liver diseases such as liver cirrhosis and even liver cancer [3,4,5]. Currently, most of the conventional drug therapies for liver fibrosis-related diseases are to remove the etiology. For example, for virus-induced liver fibrosis, priority should be given to clearing the virus, and at the same time, anti-inflammatory, liver protection, and ECM degradation-promoting treatment measures should be taken. However, etiological treatment cannot completely stop the progress of liver fibrosis [5,6]. As of 2030, the epidemiology forecast of Hepatology International shows that the mortality caused by liver cirrhosis and liver cancer is on the rise year by year and has become one of the high-risk diseases that seriously threaten human life and health. At present, although the mechanism leading to liver fibrosis has been widely understood, no effective treatment has been developed so far. For a long time, traditional Chinese medicine (TCM) has had unique advantages in preventing and treating liver fibrosis and exhibiting significant clinical efficacy [7]. TCM has multi-level, multi-channel, and multi-target pharmacological effects, such as with Curcuma, which can regulate TGF-β1/Smads and PI3K/Akt signal pathways to improve CCl_4_-induced liver fibrosis [8], while Pien-Tze-Huang alleviates CCl_4_-induced liver fibrosis through the inhibition of HSC autophagy and the TGF-β1/Smad2 pathway [9].

Saffron is a traditional dual-use natural medicine with multiple pharmacological functions planted in China, Iran, Spain, India, and other countries. Saffron can be used not only as a medicine but also as a dye, spice, and condiment, which has been used for thousands of years [10]. Saffron is the dried stigma of *Crocus sativus* L., and the yield is low. It takes about 20 flowers to obtain 1 g stigma, which is a valuable medicinal material. Saffron is rich in carotenoids, carbohydrates, proteins, anthocyanins, vitamins and minerals, as well as in crocin, picrocrocin, and safranal. Saffron is considered to be an herb with a wealth of health benefits and nutritional value in folk medicine [11,12]. Saffron has various pharmacological activities and shows great application potential in the research of many diseases. Studies have found that it has therapeutic effects on neurodegenerative diseases, depression, digestive system diseases, coronary atherosclerosis, tumors, diabetes, and eye diseases [13]. In addition, the application of saffron in liver diseases was recorded in Jingzhu Materia Medica as a classic Tibetan medicine: “Saffron can be used to treat all liver diseases”. This shows that traditional folk medicine considers saffron to have good liver-protective effects.

Currently, saffron is mainly used for treating cardiovascular diseases and liver diseases in clinics. At present, there are many preparations containing saffron (Figure 1), such as Dawa UL kurkum [14], llkl [15], saffron total glycoside tablets [16], and Liuwei xihonghua koufuye, Bawei xihonghua Qingganre capsules, which are widely used in liver diseases [17]. Modern studies have found that saffron is a good anti-inflammatory, antioxidant, and anti-apoptotic agent, and has a significant liver protective effect [17,18,19,20]. However, the mechanism of action of saffron in the treatment of liver diseases has not been particularly studied. Through traditional pharmacological evaluation, it is difficult to determine the effective mechanism of saffron with its multi-components, multi-targets, and multi-effects in the treatment of liver fibrosis. However, analysis based on network pharmacology can promote the transformation of a single drug target into a multi-component network target research mode, providing a new way to reveal the complex mechanism of TCM [21,22,23]. Therefore, we first combined network-based computational prediction with experimental verification to study the pharmacological mechanism of saffron in treating liver fibrosis. First, the main chemical spectra of saffron were identified by LC/MS. According to the structural and functional similarity of drugs, the corresponding putative target spectrum was predicted. Then, the herbal composition targets of saffron were constructed using network pharmacology, and the candidate targets of saffron against liver fibrosis were screened based on PPIs. Finally, on this basis, the molecular mechanism of the anti-liver fibrosis effect of saffron was verified by a series of experiments using a classical model of CCl_4_-induced liver fibrosis.

## 2. Result

### 2.1. Chemical Profiling of Saffron Extract by LC-MS

UPLC-MS/MS technology was used to qualitatively analyze the chemical components of saffron, and the total ion current (TIC) diagram was obtained in positive and negative ion modes (Figure 2). The data on retention time, molecular ions, major MS2 fragments, tentative compound assignments, and molecular formulae are shown in Table 1. A total of 56 compounds were identified by comparison of mass spectrometry data with published data, including flavonoids, amino acids, carotenoids, phenolics, and organic acids. As seen in Table 1, seven amino acids were detected from saffron, namely DL-arginine, isoleucine, L-glutamic acid, L-phenylalanine, L-threonic acid, proline, valine; three carotenoid classes, crocin I, crocin II, and crocetin; four flavonoid components, namely kaempferol, kaempferol-7-O-glucoside, trifolin, and rutin; as well as carvone, safranal, and monosaccharide and polysaccharide components.

### 2.2. Network Pharmacological Analysis of Saffron-Alleviated Liver Fibrosis

#### 2.2.1. Screening the Main Active Components in Saffron

According to the standard compounds, existing literature data, and obtained MS data, a total of 56 chemical components were identified. Based on the ADME properties of the components and their content in saffron, we identified nine components that can be used as the representative components of subsequent network pharmacology for analysis (Table 2).

#### 2.2.2. Putative Targets of Saffron for the Treatment of Liver Fibrosis

Nine active ingredients in saffron were identified through screening, and 435 potential targets of nine active ingredients in saffron were obtained based on database screening. We performed a Venn analysis on 6134 candidate targets related to liver fibrosis in the database and potential targets of active ingredients in saffron, of which 321 overlapped (Figure 3A). We hypothesized that these 321 targets were the putative targets for saffron in the treatment of liver fibrosis and constructed a network of herbal ingredient targets (Figure 3B), with a total of 331 nodes (nine active ingredients and 321 putative targets) and 682 edges.

#### 2.2.3. Construction of the PPIs Network and Network Analysis

A total of 321 potential targets of saffron were imported into the string database and Cytoscape 3.7.1 to obtain the PPIs network (Figure 4A). The PPIs network showed 255 nodes and 1986 edges, with an average node degree of 7.86 and an average local clustering coefficient of 0.57, indicating the interaction between protein and function. The results also show that SRC, HSP90AA1, MAPK1, MAPK3, TP53, STAT3, AKT1, and other targets are located at the core of the network and play key regulatory roles in the PPIs network. We obtained the top 30 core genes of the PPIs network according to the degree value and visualized them as histograms using R language (Figure 4B).

KEGG and GO enrichment analyses were performed on 255 targets obtained by PPI analysis. The biological processes involved in target enrichment are mainly related to chemical–organic substance reactions. In terms of the enriched molecular functions, the target proteins were mainly related to protein binding, signal transduction activity, and macromolecular complex binding (Figure 5A). This suggested that the active components of saffron may exert antifibrotic effects by participating in various bioregulatory processes. Subsequently, pathway enrichment analysis based on the KEGG pathway database revealed that the putative targets of saffron for the treatment of hepatic fibrosis are significantly involved in the calcium signaling pathway, the HIF-1 signaling pathway, endocrine resistance, the PI3K/Akt signaling pathway, lipid and atherosclerosis, and the cAMP signaling pathway (Figure 5B,C). These pathways involve inflammation, the endocrine system, the signaling system, and the cell cycle. In particular, there are two pathways closely related to the process of liver fibrosis, including the HIF-1 and PI3K/Akt signaling pathways. Based on these predictions, we selected PIK3CA, AKT1, HIF1A, VEGFA, and other related targets as the candidate targets of saffron for anti-hepatic fibrosis and conducted further experimental validation.

### 2.3. Experimental Validations of the Pharmacological Effects and Molecular Mechanisms of Saffron-Alleviated Liver Fibrosis

#### 2.3.1. Effects of Saffron Extract on Liver Index and Spleen Index of Liver Fibrosis Mice

In general, body weight, liver, and spleen weights were recorded, and liver and spleen indices were calculated. Compared with the control group, liver weight and liver index, spleen weight, and spleen index were significantly increased in the model group. In contrast, compared with the model group, the saffron treatment group could effectively reduce spleen index (−38.6% and −48.2%) (*p* < 0.01) and liver index (−17.8% and −27.6%) (*p* < 0.05 and *p* < 0.01) (Figure 6).

#### 2.3.2. Saffron Extract Ameliorated the Hepatic Injury Markers in Liver Fibrosis Mice

In order to investigate the changes in liver function after saffron treatment, the levels of serum ALT, AST, and TBIL were measured. Compared with the control group, the levels of serum ALT, AST, and TBIL in the model group increased significantly, indicating that CCl_4_ treatment leads to serious liver injury. Importantly, saffron treatment significantly reduced serum AST (−33.1% and −34.1%) (*p* < 0.01), ALT (−49.9% and −63.4%) (*p <* 0.001), and TBIL (−19.7% and −22.1%) (*p* < 0.05) levels in a dose-dependent manner, indicating that saffron significantly reduces CCl_4_-induced hepatotoxicity (Figure 7).

#### 2.3.3. Saffron Extract Alleviated Liver Histopathological Changes in Liver Fibrosis Mice

The livers of mice in each group were visually observed and compared (Figure 8A). The appearance of the liver in the control group was smooth, reddish brown, soft, and elastic. The appearance of the liver in the model group was brownish yellow, hard, and blunt, the liver edge was thick, and many small gray-white nodules were distributed on the surface. In addition, some surfaces had slight orange peel changes. Compared with the model group, after saffron treatment, the above pathological characteristics were significantly reduced.

The histopathological changes induced by CCl_4_ treatment and saffron treatment were studied by H&E staining. The liver of the control group showed that the hepatocyte structure was complete, the cells were arranged closely and regularly, and the nuclear circle was in the middle. CCl_4_ treatment caused severe pathological changes, including excessive inflammatory infiltration, necrosis, and fibrous tissue proliferation of the liver. Most importantly, saffron significantly alleviated CCl_4_-induced liver inflammation and fibrosis (Figure 8B).

Collagen deposition in the liver was detected by picrosirius red staining. CCl_4_ treatment caused fiber expansion in some portal and central venous regions. In addition, compared with the control group, the Sirius red positive area in the CCl_4_ treatment group was significantly increased. In contrast, saffron treatment significantly reduced collagen deposition in the liver (Figure 8D).

Masson’s trichrome staining can be used to evaluate collagen accumulation in liver tissues (Figure 8C). The results show that, in the control group, only a small amount of collagen (blue area) could be seen in the main vascular wall of the liver tissue, the hepatocyte cords were arranged regularly, the cell structure was complete, there was no fat vacuole formation in the hepatocytes, and there was no visible collagen fiber deposition in the portal area. In the model group, there were a large number of thick and proliferating collagen fibers between the hilar region and the lobule, and the normal structure of the lobule was destroyed. In contrast, the collagen fiber deposition, pseudo lobule formation, and inflammatory cell infiltration in the liver tissue of saffron-treated rats were improved to some extent. These histopathological results show that saffron treatment significantly improved the pathological characteristics of liver fibrosis and had a protective effect on liver tissue.

#### 2.3.4. Saffron Extract Attenuated Liver Damage and Collagen Deposition in Liver Fibrosis Mice

During liver fibrosis, activated hematopoietic stem cells lead to the proliferation and overexpression of α-SMA, and Col II is an important initiator of liver fibrosis. The results show that α-SMA and Col II were minimally expressed in normal liver tissues. Compared with the control group, the positive expression area of α-SMA and Col II increased significantly (*p* < 0.01), and the brownish-yellow area became darker, mainly distributed in the portal vein area and the fiber area, and the degree of hepatic fibrosis deepened in the model group. Compared with the model group, the positive expression area of α-SMA and Col II in the liver tissue of the saffron group were significantly reduced (*p* < 0.01), and the brown-yellow area was reduced in area and lightened in color (Figure 9). Taken together, these findings suggest that saffron extract can inhibit the accumulation of collagen and alleviate and reverse the production of liver fibrosis.

#### 2.3.5. Saffron Extract Inhibited the AKT/HIF-1α/VEGF Signaling Pathways of Liver Fibrosis Mice

The mRNA and protein expression levels of the genes involved in the AKT/HIF-1/VEGF signaling pathway obtained from the network pharmacological screen were determined by RT-PCR and Western blotting. RT-PCR was used to detect the mRNA expression levels of AKT/HIF-1α/VEGF signaling pathway-related genes (Figure 10). The results show that the mRNA expression of AKT1, PIK3CA, VEGFA, HIFA, EGFR, PRKCA, FLT1, and EGLN1 in the liver AKT/HIF-1/VEGF signaling pathway in the model group was significantly lower than that in the control group (*p* < 0.001). The mRNA levels of the saffron-treated group were significantly lower than those of the model group.

Compared to the model group, the saffron group decreased the expression ratio of p-AKT/AKT. This is a downstream molecule of PI3K that activates the VEGF/HIFA signaling pathway. The result suggests that saffron affected the expression of p-AKT in fibrotic liver tissue. This result demonstrates that saffron has anti-inflammatory activity by inhibiting the PI3K/AKT-mediated VEGF/HIFA signaling pathway, thereby reducing the release of cytokines and thus attenuating the inflammatory response in the liver fibrosis process and inhibiting pathological angiogenesis (Figure 11).

## 3. Discussion

Liver fibrosis is a complex disease, and current treatment options for liver fibrosis remain limited. The research results prove that saffron has a therapeutic effect on liver fibrosis, and its mechanism of action is related to the regulation of the AKT/HIF-1α/VEGF signaling pathway. In the CCl_4_-induced liver fibrosis model, saffron inhibited the inflammatory infiltration in the process of liver fibrosis by regulating and inhibiting the inflammatory AKT/PI3K signaling pathway, while regulating HIF-1α/VEGF signaling pathway can improve the insufficient blood supply of hepatocytes, vascular diseases, and cell hypoxia caused by the destruction of liver tissue structure, as shown in Figure 12.

The existence of long-term inflammatory reactions and continuous hepatocyte damage are the main factors that can induce liver fibrosis. Through network pharmacology, we screened that Akt/PI3K signaling pathway is the main inflammatory pathway of saffron to improve liver fibrosis. Existing studies have shown that saffron can treat liver diseases through its anti-inflammatory effect. Safranal, the main bioactive component of saffron, can significantly inhibit the proliferation of liver cancer cells and induce apoptosis, showing anti-inflammatory properties, and having significant inhibition effects on inflammatory markers such as NF-κB, COX2, iNOS, and TNF-α [24]. Some studies have also shown that crocetin is a potential bioactive component of saffron in the treatment of nonalcoholic fatty liver disease, and its mechanism of action includes inhibiting oxidative stress, alleviating inflammation, and upregulating the expression of Nrf2 and HO-1 [25].

Liver fibrosis and cirrhosis belong to the category of blood stasis in TCM. The methods of treating liver fibrosis in TCM mainly focus on herbs of promoting blood circulation and removing blood stasis [26]. Hepatic sinusoidal capillarization characterized by pathological angiogenesis is an important pathological manifestation in the process of liver fibrosis [25,27,28]. Studies have shown that HIF-1α and VEGF are thought to play key roles in angiogenesis and remodeling. VEGF promotes vascular permeability, extracellular matrix degeneration, and vascular endothelial cell migration [29,30]. HIF-1α is a key transcription factor in response to hypoxic stress, and its expression is significantly increased in liver fibrotic tissues [31,32]. In TCM, activating blood circulation and removing blood stasis are the main effects of saffron. Previous studies have also shown that saffron treats diseases through the HIF-1α/VEGF signaling pathway, and crocin inhibits angiogenesis and metastasis of colorectal cancer cells by targeting NF-κB and blocking the TNF-α/NF-κB/VEGF pathway [33]. Crocetin and its Glycoside Crocin modulate VEGFR2/SRC/FAK and VEGFR2/MEK/ERK, thereby inhibiting angiogenesis to varying degrees [34]. Safranin attenuates atherosclerosis by modulating the expression of eNOS and HIF-1α in lipoprotein mice [35,36].

## 4. Materials and Methods

### 4.1. Chemicals and Reagents

ALT, AST, and TBIL kits were purchased from Wuhan Servicebio Biotechnology Co., Ltd. (Wuhan, China). Antibody human α-SMA and Col II are from Abcam (Cambridge, MA, USA).

### 4.2. Preparation of Saffron Extract

Saffron was purchased from Sichuan NEAUTUS Traditional Chinese Medicine Co., Ltd. (Chengdu, China). We prepared saffron sample powder by grinding accurately weighed 80 mg of saffron powder, placing it in a 50 mL brown centrifuge tube, adding 20 mL of pure water, and extracting it by ultrasound in 25 °C water for 30 min. After centrifuging the extract at 14,000 r/min for 10 min, we took the supernatant and put the saffron sample away for storage at −4 °C for future use.

### 4.3. LC-MS Analysis Conditions

Chromatographic conditions of saffron extract: mobile phase A (0.1% formic acid water)—mobile phase B (acetonitrile), elution gradient 0~5 min, 95~70% A; 5~25 min, 70~20% A; Flow rate 0.2 mL/min; Column temperature 30 °C, injection volume 3 μL.

Mass spectrometry conditions: capillary voltage 2.5 kV; source temperature 120 °C; desolvent gas temperature 450 °C; desolvent gas flow rate 600 L/h; air flow rate in conical hole 50 L/h; spray 6.5 bar; source offset ion scanning range *m*/*z* 50–1200 Da; scanning time 0.2 s.

### 4.4. Network Construction and Analysis

The active chemical constituents of saffron extracts were obtained by LC/MS analysis combined with TCMSP (https://tcmspw.com (accessed on 28 May 2022)). Next, we searched PubChem (https://pubchem.ncbi.nlm.nih.gov (accessed on 30 May 2022)), Drugbank (https://www.drugbank.ca/drugs (accessed on 2 June 2022)), and used a molecular similarity matching tool based on Swisstargetprediction (http://www.swisstargetprediction.ch (accessed on 2 June 2022)) and a molecular similarity matching tool based on TCMSP to identify potential target proteins of saffron extracts. Then, through the Genecards server (https://www.genecards.org (accessed on 2 June 2022)), the related targets of liver fibrosis were determined by the Online Mendelian genetic man (OMIM) database. Based on the previous steps, two groups of targets, drug-related genes, and disease targets were prepared. Using the Venn map software (http://bioinformatics.psb.ugent.be/webtools/Venn/ (accessed on 2 June 2022)), cross genes were screened by R software (version 3.5.1). According to the interaction data, we constructed the interaction network of the active components and hypothetical targets of saffron extract in the treatment of liver fibrosis and visualized the PPIs network with Cytoscape software (version 3.7.1). The network analyzer plug-in and cytonca plug-in Cytoscape were used to measure the topological importance of nodes and networks. Finally, using the Bioconductor software package (version 3.17), including the Panther classification system, bioinformatics annotation evaluation was performed on proteins with overlapping expression patterns using R software (version 3.5.1), GO annotation database website (http://geneontology.org (accessed on 10 June 2022)), and KEGG pathway enrichment analysis (https://www.kegg.jp (accessed on 12 June 2022)).

### 4.5. Animal Study Design

Male KM mice (8 weeks old) were purchased from Institute of Laboratory Animals, Chongqing Institute of Traditional Chinese Medicine, Animal License No. SCXK (Chongqing, China) 2017-0003. This study was reviewed and approved by the Animal Ethics Committee of Chengdu University of Traditional Chinese Medicine (2014DL-023). The animals were acclimatized to the environment for 1 week before the experiment. Mice were housed in an environment with controlled temperature (22 ± 2 °C) and humidity (60 ± 10%), with a light/dark cycle of 12 h, and fed with standard chow. Mice were randomly divided into four groups (N = 8): control, model, saffron high-dose group (100 mg/kg), and saffron low-dose group (50 mg/kg). A mouse model of hepatic fibrosis was prepared by intraperitoneal injection of CCl_4_ dissolved in olive oil (twice a week) for 6 weeks. Meanwhile, the saffron group was gavaged with saffron extract once a day, and the control and model groups were gavaged with equal volume of 0.9% saline for 6 weeks.

### 4.6. Organ Coefficient

At the end of the experiment, the weight of the liver and spleen were recorded. Calculation of liver coefficient and spleen coefficient (organ coefficient (%) = organ (g)/body weight (g) × 100%) was conducted.

### 4.7. Biochemical Analysis

Blood samples were collected at the end of the study. The serum was centrifuged at 3000× *g* for 15 min and subjected to biochemical analysis. The liver function indices (ALT, AST, and TBIL) were measured by ELISA kit.

### 4.8. HE Staining, Sirius Red Staining, and Masson Staining

Livers were fixed with 4% paraformaldehyde for 48 h, embedded in paraffin, cut into 5 µm thick sections and stained with hematoxylin eosin. The slices were washed twice in acidified water and then dehydrated with 100% ethanol. The slices were immersed in Masson A solution overnight, and then washed with tap water slightly; the excess water was slightly drained. The slices were then immersed in Masson D solution for 6 min and the excess water was drained as much as possible. The slices were then treated with Masson E solution for about 1 min, they were not washed, and the excess Masson E solution was drained slightly. The slices were then directly dyed in Masson F solution for 2–10 s, and then dehydrated with 100% ethanol. The positive staining area was analyzed and calculated by the Image-Pro Plus 6.0 system.

### 4.9. Immunohistochemistry

Paraffin-embedded liver tissue sections of 5 µm thickness were subjected to immunohistochemistry, deproteinization, and 0.3% hydrogen peroxide treatment for 15 min to block endogenous peroxidase activity. Liver sections were further blocked with 2% bovine serum albumin and then reacted with rabbit α-SMA and Col II and incubated at 4 °C for 16 h. After washing, the sections were incubated with BSA for 60 min at room temperature. Samples were analyzed by light microscopy and α-integrated optical density (IOD) values expressed as α-SMA and Col II were detected using Image pro + 6.0 software.

### 4.10. Western Blotting Analysis

The protein samples were loaded into 10% sodium dodecyl sulfate polyacrylamide gel electrophoresis (SDS-PAGE) and then transferred to the PVDF membrane. Membrane culture in QuickBlock™ buffer (beyotime Biotechnology, Shanghai, China) was blocked at room temperature for 1 h and then incubated with primary antibody AKT1, VEGFA, HIF1A, PI3K, P-AKT, and P-PI3K from Affinity (Shanghai, China) at 4 °C overnight. The membrane was then incubated with HRP binding secondary antibodies at room temperature for 1 h. The bands were observed using the chemiluminescence kit.

### 4.11. Real-Time Quantitative PCR Assay

Total RNA was extracted from liver samples by Trizol* Reagent and converted into complementary DNA using qscriptcdna super mixing reagent. Using platinum according to the manufacturer’s instructions, *SYBR* Green q-PCR was used to detect the gene expression at the target gene level. The PCR primer sequences used in this study are shown in Table 3.

### 4.12. Statistical Analysis

All data were expressed as mean ± SEM, and one-way analysis of variance (ANOVA) was performed, followed by the Bonferroni posttest. *p* < 0.05 in different groups was considered significant.

## 5. Conclusions

In conclusion, our results provide new evidence that saffron ameliorates inflammatory infiltration in liver fiber processes, as well as vasculopathy and cellular hypoxia in a model of fibrosis through AKT/HIF-1α/VEGF signaling. Our study shows that saffron can be used as a candidate drug and health food for the prevention and treatment of liver fibrosis.

## Figures and Tables

**Figure 1 molecules-28-07238-f001:**
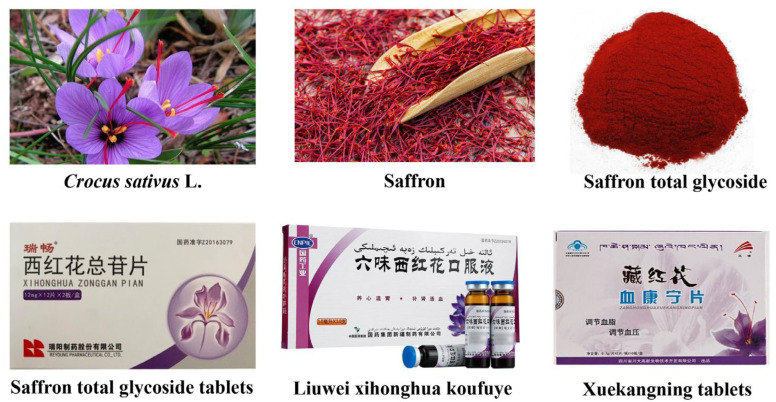
Saffron and its Chinese patent medicine for liver disease.

**Figure 2 molecules-28-07238-f002:**
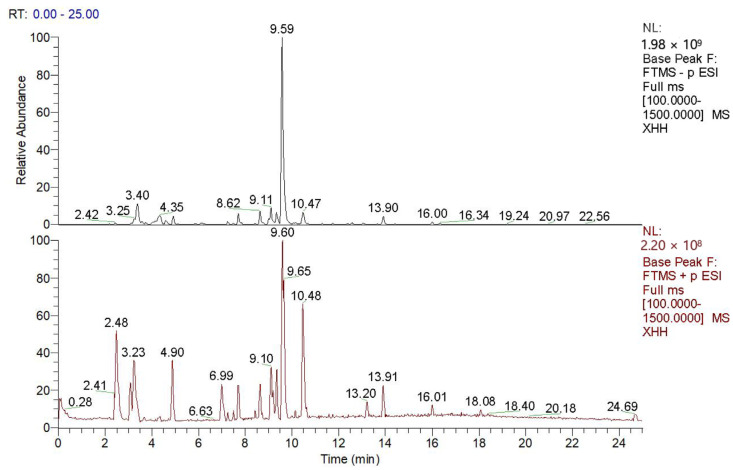
Base peak ion flow diagram of positive ion and negative ion of saffron.

**Figure 3 molecules-28-07238-f003:**
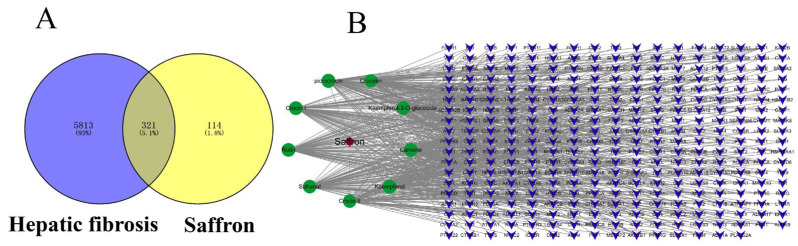
Network construction. (**A**) Venn diagram of saffron and liver fibrosis candidate targets. (**B**) Herbal component target network of the anti-fibrosis effect of saffron. The red node represents herbal medicine, and the blue nodes represent 321 putative targets of saffron in the treatment of liver fibrosis. The green nodes represent 9 active components in saffron.

**Figure 4 molecules-28-07238-f004:**
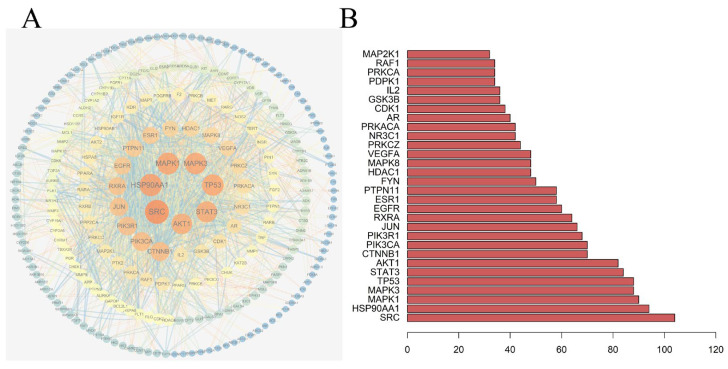
PPIs network of interaction targets. (**A**) PPIs network; (**B**) Core targets histogram.

**Figure 5 molecules-28-07238-f005:**
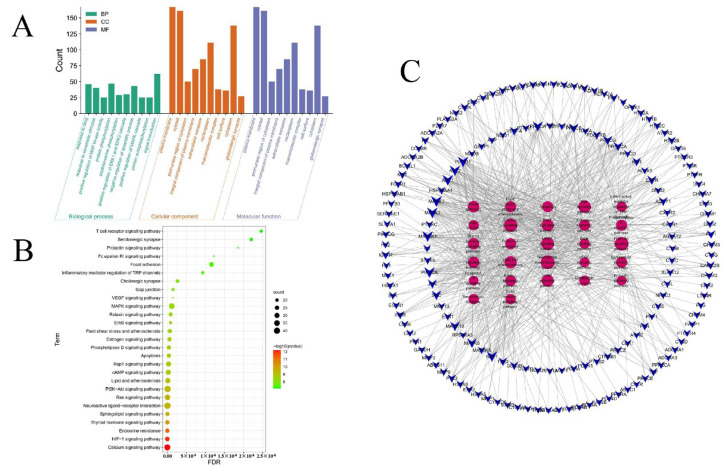
Functional analysis. (**A**) GO enrichment analysis of putative targets. (**B**) KEGG pathway enrichment analysis of hypothetical targets. (**C**) Target-signaling pathway network. The red node represents the main signal pathway, and the blue node represents the putative target of saffron in the treatment of liver fibrosis.

**Figure 6 molecules-28-07238-f006:**
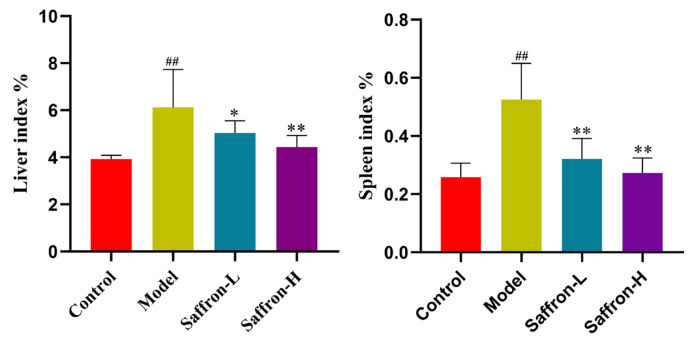
The effect of saffron on organ index, ^##^
*p* < 0.01 (VS. control group), * *p* < 0.05 (VS. model group), and ** *p* < 0.01 (VS. model group).

**Figure 7 molecules-28-07238-f007:**
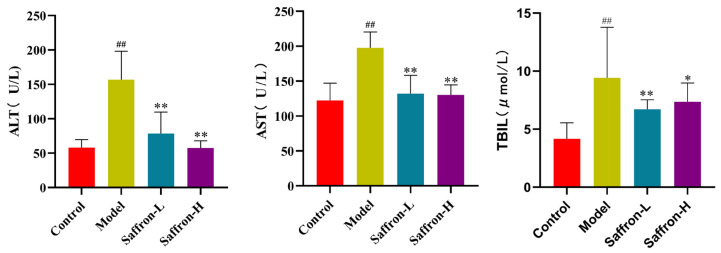
Effect of saffron on serum biochemical parameters of CCl_4_ mice, ^##^
*p* < 0.01 (VS. control group), * *p* < 0.05 (VS. control group), ** *p* < 0.01 (VS. control group).

**Figure 8 molecules-28-07238-f008:**
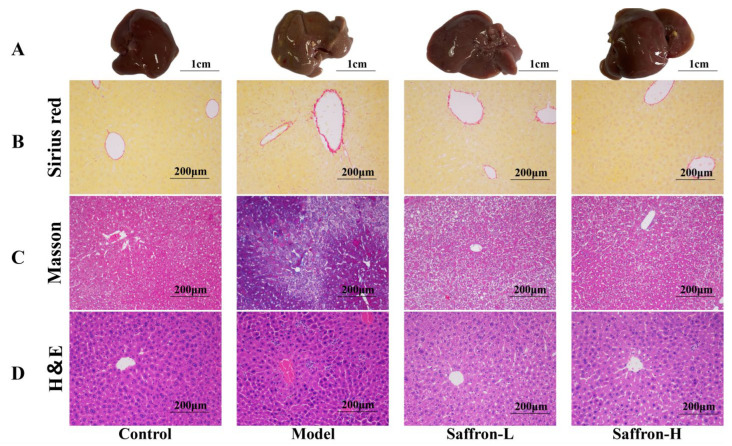
Effect of saffron on histological changes of CCl_4_ mice. (**A**) Representative photos of liver tissue. (**B**) Sirius red staining of liver tissue (scale bar, 200 μm). (**C**) Masson staining of liver tissue (scale bar, 200 μm). (**D**) H&E staining of liver tissue (scale bar, 200 μm).

**Figure 9 molecules-28-07238-f009:**
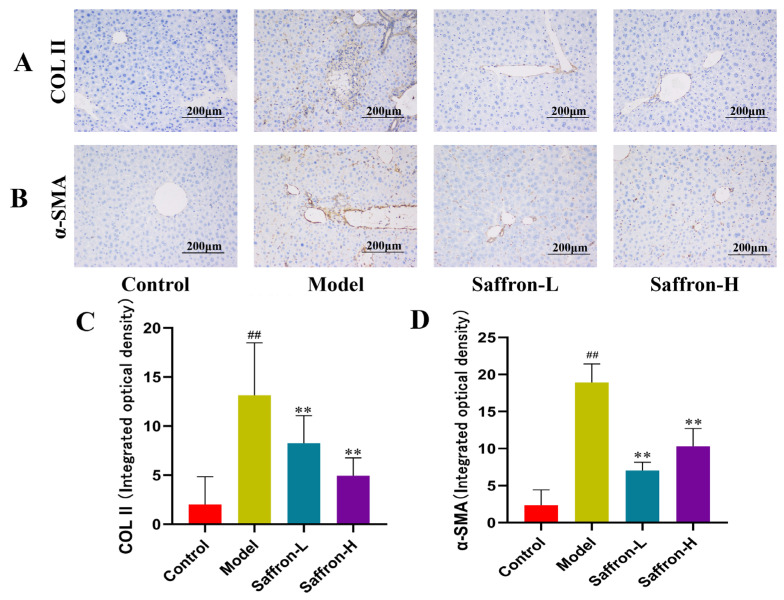
Effect of saffron on liver α-SMA and Col II expression in CCl_4_ mice. (**A**) The effect of saffron on liver α-SMA and Col II expression as detected by immunohistochemistry. (**B**) Quantitative analysis of the effect of saffron on the expression of α-SMA and Col II. (**C**) Staining intensity of the α-SMA protein. (**D**) Staining intensity of the Col II protein. Data are presented as means ± SD (*n* = 3 in each group). ^##^
*p* < 0.01 (vs. control), ** *p* < 0.01 (vs. model).

**Figure 10 molecules-28-07238-f010:**
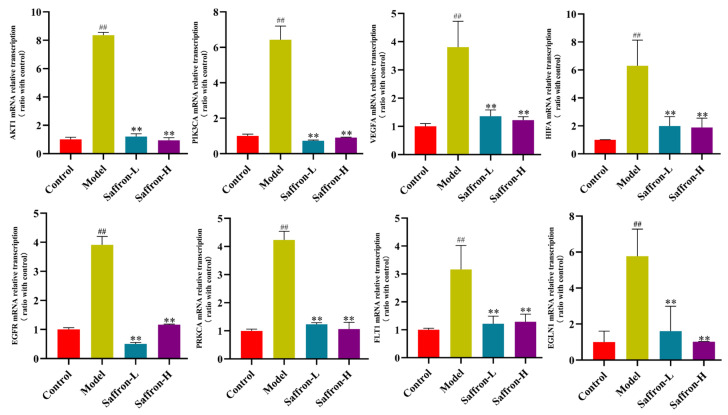
mRNA levels of AKT/HIF-1α/VEGF signaling pathway-related genes in liver tissue. Data are presented as means ± SD (*n* = 3 in each group). ^##^
*p* < 0.01 (vs. control), ** *p* < 0.01 (vs. model).

**Figure 11 molecules-28-07238-f011:**
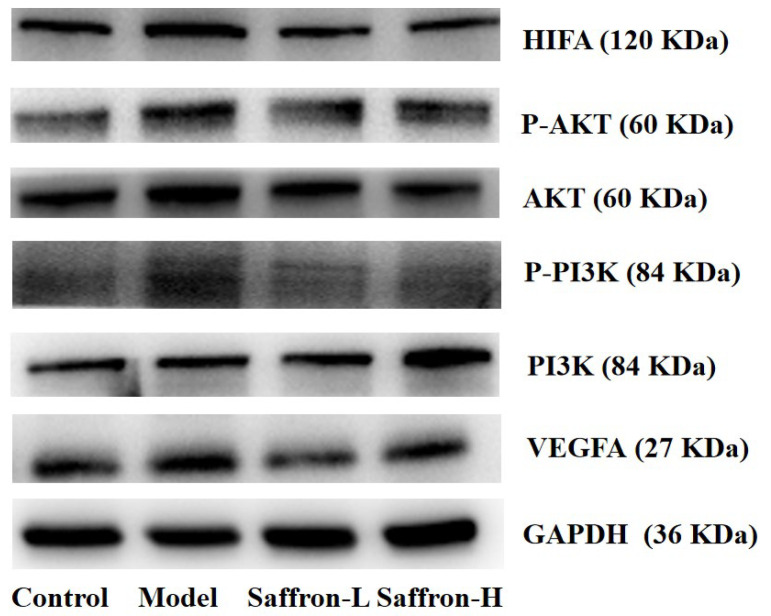
Western blot assay for determining the expression of AKT, PI3K, VEGF, HIFIA, P-AKT, and P-PI3K expression in liver tissue. ^##^ *p* < 0.01 (VS. control group), * *p* < 0.05 (VS. control group), ** *p* < 0.01 (VS. control group).

**Figure 12 molecules-28-07238-f012:**
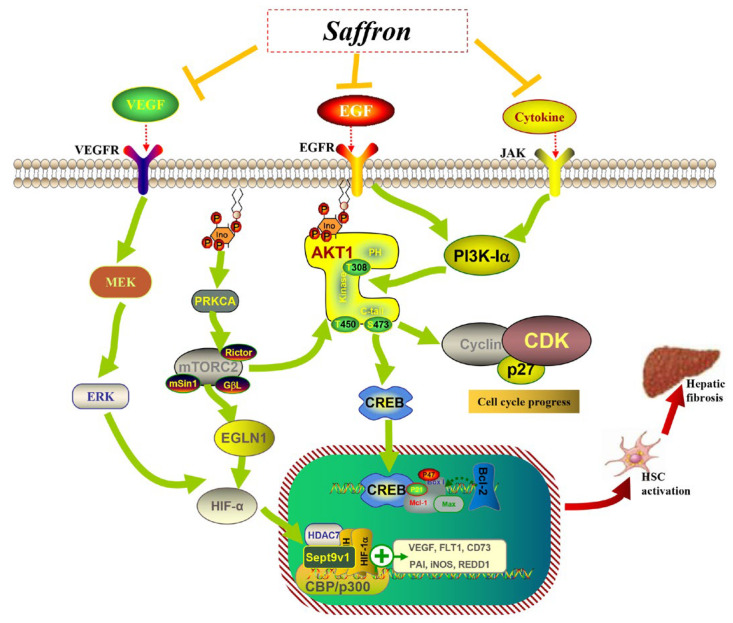
Mechanism of action of saffron against liver fibrosis.

**Table 1 molecules-28-07238-t001:** Molecular formula-related data of compounds in saffron.

No.	RT (min)	Ionization Mode	Experimental *m*/*z*	Molecular	ppm	Tentative Compound	Molecular Formula	MS/MS Fragments
1	2.384	[M + H]^+^	175.119	175.119	0.57	DL-Arginine	C_6_H_14_N_4_O_2_	175, 151, 116, 70
2	2.490	[M + H]^+^	104.107	104.107	3.84	Choline	C_5_H_13_NO	104, 86, 60
3	3.099	[M − H]^−^	146.045	146.044	4.72	L-Glutamic acid	C_5_H_9_NO_4_	146, 128, 102
4	3.167	[M + H]^+^	184.073	184.073	0.00	Phosphocholine	C_5_H_14_NO_4_P	184, 124, 86
5	3.196	[M + H]^+^	236.110	236.108	8.05	3-methyl-5-oxo-5-(4-toluidino) pentanoic acid	C_13_H_17_NO_3_	258, 184, 124, 104
6	3.242	[M + H]^+^	138.055	138.055	0.00	Trigonelline	C_7_H_7_NO_2_	138, 120, 110, 92, 78
7	3.252	[M + H]^+^	360.150	360.152	−7.25	α-Lactose	C_12_H_22_O_11_	316, 163, 145, 127, 97, 85
8	3.270	[M + H]^+^	116.071	116.072	−6.46	Proline	C_5_H_9_NO_2_	116, 93, 70
9	3.277	[M − H]^−^	179.055	179.056	−2.07	Glucose	C_6_H_12_O_6_	129, 89, 71, 59
10	3.281	[M + FA − H]^−^	549.167	549.169	−3.17	D-Raffinose	C_18_H_32_O_16_	503, 346, 113, 89, 71
11	3.320	[M + H]^+^	365.105	365.106	−2.16	D-(+)-Maltose	C_12_H_22_O_11_	365, 203, 185, 98, 69
12	3.386	[M+FA-H]^−^	342.116	342.116	−0.50	Sucrose	C_12_H_22_O_11_	341, 179, 119, 101, 89, 71
13	3.395	[M − H]^−^	195.050	195.051	−1.85	Gluconic acid	C_6_H_12_O_7_	195, 177, 129, 75
14	3.436	[M + H]^+^	118.086	118.086	2.03	Valine	C_5_H_11_NO_2_	118, 95, 72
15	3.731	[M − H]^−^	135.029	135.029	−2.07	L-threonic acid	C_4_H_8_O_5_	135, 89, 75
16	4.144	[M + H]^+^	123.056	123.056	−5.85	Nicotinamide	C_6_H_6_N_2_O	123, 96, 80
17	4.209	[M – H]^−^	175.024	175.024	−2.00	Ascorbic acid	C_6_H_8_O_6_	175, 115, 87, 71
18	4.225	[M + H]^+^	348.070	348.071	−2.13	Adenosine 5′-monophosphate	C_10_H_1_N_5_O_7_P	348, 326, 136
19	4.333	[M − H + HAc]^−^	179.055	179.056	−2.18	D-Fructose	C_6_H_12_O_6_	179, 119, 113, 89, 71, 59
20	4.398	[M − H]^−^	193.035	193.035	−1.45	Galacturonic acid	C_6_H_10_O_7_	193, 103, 71, 59
21	4.845	[M − H]^−^	133.013	133.014	−2.33	DL-Malic acid	C_4_H_6_O_5_	133, 115, 71
22	4.876	[M − H]^−^	268.104	268.105	−2.87	Adenosine	C_10_H_13_N_5_O_4_	268, 136
23	4.878	[M + H]^+^	136.062	136.063	−6.39	Adenine	C_5_H_5_N_5_	136, 91, 72, 55
24	4.905	[M + H]^+^	132.102	132.103	−5.45	Isoleucine	C_6_H_13_NO_2_	132, 113, 108, 90, 86, 72
25	5.106	[M + H]^+^	113.035	113.036	−6.81	Uracil	C_4_H_4_N_2_O_2_	113, 96, 70
26	5.107	[M − H]^−^	243.062	243.062	−1.56	Uridine	C_9_H_12_N_2_O_6_	243, 200, 153, 110, 82
27	5.544	[M − H]^−^	259.022	259.023	−1.31	Glucose 1-phosphate	C_6_H_13_O_9_P	259, 215, 96, 78
28	5.675	[M − H]^−^	171.006	171.006	−1.99	Glycerol 3-phosphate	C_3_H_9_O_6_P	171, 124, 96, 78
29	5.715	[M + H]^+^	152.057	152.057	−5.20	Guanine	C_5_H_5_N_5_O	152, 143, 134, 109, 96
30	5.718	[M + H]^+^	284.099	284.100	−3.48	Guanosine	C_10_H_13_N_5_O_5_	152, 135, 110
31	5.869	[M − H]^−^	191.019	191.019	−1.88	Citric acid	C_6_H_8_O_7_	191, 147, 111, 102, 97, 85
32	6.259	[M − H]^−^	117.018	117.019	−2.56	Succinic acid	C_4_H_6_O_4_	117, 99, 73, 71
33	6.587	[M − H]^−^	323.029	323.030	−3.93	Uridine monophosphate	C_9_H_13_N_2_O_9_P	323, 211, 150, 138, 111, 96
34	6.622	[M + H]^+^	166.086	166.086	1.14	L-Phenylalanine	C_9_H_11_NO_2_	166, 138, 120, 103, 91, 74
35	7.250	[M + H]^+^	449.108	449.108	−1.78	Kaempferol-7-O-glucoside	C_21_H_20_O_11_	449, 287, 231, 183, 160, 137
36	7.712	[M − H]^−^	179.055	179.056	−1.62	Fructose	C_6_H_12_O_6_	134, 119, 113, 101, 89, 71
37	8.085	[M + H]^+^	346.162	346.163	−1.99	Jasminoside B	C_16_H_26_O_8_	238, 185, 167, 139, 121
38	8.734	[M + H]^+^	167.107	167.107	−4.49	6-Pentyl-2H-pyran-2-one	C_10_H_14_O_2_	167, 149, 143, 125, 121, 107
39	9.015	[M − H]^−^	179.055	179.056	−1.62	9-Fluorenone	C_13_H_8_O	178, 134, 119, 113, 101, 89
40	9.340	[M − H]^−^	609.146	609.143	4.81	Rutin	C_27_H_30_O_16_	609, 284, 255, 227
41	9.410	[M − H]^−^	145.050	145.050	−2.28	3-Methylglutaric acid	C_6_H_10_O_4_	145, 121, 111, 101
42	9.443	[M + H]^+^	169.122	169.123	−3.07	10-HAD	C_10_H_18_O_3_	169, 151, 123, 109, 95, 81
43	9.564	[M + H − H_2_O]^+^	135.080	135.081	−3.33	2,4-Dimethylbenzaldehyde	C_9_H_10_O	135, 119, 107, 93, 91, 79
44	9.566	[M + H]^+^	365.120	365.121	−2.08	Coniferin	C_16_H_22_O_8_	365, 337, 206, 187
45	9.587	[M + H]^+^	123.117	123.118	−5.28	1,2,3,4-Tetramethyl-1,3-cyclopentadiene	C_9_H_14_	123, 95, 81, 67
46	9.647	[M − 2H]^2−^	975.371	975.371	−0.62	Crocin I	C_44_H_64_O_24_	651, 327, 283
47	10.149	[M + H]^+^	369.131	369.132	−2.38	Gibberellic acid	C_19_H_22_O_6_	369, 66, 61
48	10.465	[M − 2H]^2−^	813.319	813.319	−0.98	Crocin II	C_38_H_54_O_19_	651, 489, 327
49	10.581	[M − H]^−^	447.094	447.095	−2.26	Trifolin	C_21_H_20_O_11_	447, 380, 284, 255, 227
50	13.047	[M + H]^+^	329.175	329.176	−4.31	Crocetin	C_20_H_24_O_4_	329, 311, 293, 265, 197
51	13.204	[M + H]^+^	151.112	151.112	−4.96	Safranal	C_10_H_14_O	151, 133, 123, 81, 67
52	15.288	[M − H]^−^	327.218	327.219	−3.97	Corchorifatty acid F	C_18_H_32_O_5_	327, 291, 229, 211, 171
53	18.076	[M + H]^+^	139.112	139.112	−5.61	Isophorone	C_9_H_14_O	139, 121, 110, 97, 81, 69
54	18.641	[M + H]^+^	165.091	165.092	−4.66	1-(4-methoxyphenyl) propane-1,2-diol	C_10_H_14_O_3_	165, 121, 119, 105, 91
55	19.891	[M + H]^+^	287.055	287.055	−2.61	Kaempferol	C_15_H_10_O_6_	287, 259, 231, 185
56	24.689	[M + H]^+^	151.112	151.112	−4.24	Carvone	C_10_H_14_O	151, 133, 123, 81

**Table 2 molecules-28-07238-t002:** Detailed information on qualitative analysis of chemical components in saffron by LC-MS.

NO.	Chemical Name	CAS NO	Compound CID	2D Structure	Formula	MolecularWeight	OB%	DL	Caco-2
1	Crocetin	27876-94-4	5281232	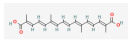	C_20_H_24_O_4_	328.40	35.30	0.26	0.54
2	Carvone	99-49-0	7439	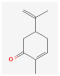	C_10_H_14_O	150.10	49.47	0.03	1.35
3	Kaempferol	520-18-3	5280863	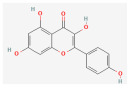	C_15_H_10_O_6_	286.05	41.88	0.26	0.24
4	Rutin	153-18-4	5280805	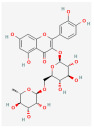	C_27_H_30_O_16_	610.15	3.20	0.68	−1.93
5	Crocin Ι	42553-65-1	5281233	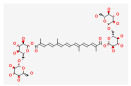	C_44_H_64_O_24_	976.96	2.54	0.12	−4.23
6	Crocin ΙΙ	55750-84-0	132399078	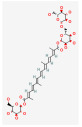	C_38_H_54_O_19_	814.82	1.65	0.21	−3.48
7	Kaempferol-7-glucoside	480-10-4	5282102	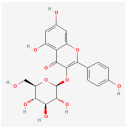	C_21_H_20_O_11_	448.10	14.03	0.74	−1.34
8	Picrocrocin	138-55-6	130796	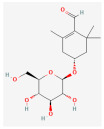	C_16_H_26_O_7_	330.37	33.71	0.04	0.69
9	Safranal	116-26-7	61041	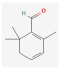	C_10_H_14_O	150.22	39.56	0.04	1.39

**Table 3 molecules-28-07238-t003:** Gene primer sequence.

Gene	Forward Sequence (5′-3′)	Reverse Sequence (5′-3′)
AKT1	ATGAACGACGTAGCCATTGTG	TTGTAGCCAATAAAGGTGCCAT
VEGFA	GCACATAGAGAGAATGAGCTTCC	CTCCGCTCTGAACAAGGCT
HIF1A	ACCTTCATCGGAAACTCCAAAG	ACTGTTAGGCTCAGGTGAACT
PIK3CA	CCACGACCATCTTCGGGTG	ACGGAGGCATTCTAAAGTCACTA
EGFR	GCCATCTGGGCCAAAGATACC	GTCTTCGCATGAATAGGCCAAT
PRKCA	TTGTCCAAGGAAGCCGTCTC	CCTTTGCCACACACTTTGGG
EGLN1	CTGGAGTACATCGTGCCG	GCCGTTTATCCTGTAGTTGC
FLT1	GACTGGTGAGGATAGCTCTACT	ATCCAATCCCTGGCCAGTC
GAPDH	GGCCTTCCGTGTTCCTACC	TGCCTGCTTCACCACCTTC

## Data Availability

The data that support the findings of this study are available upon reasonable request.

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
