# Peer review of "Hepatoprotective Effect of Medicine Food Homology Flower Saffron against CCl_4_-Induced Liver Fibrosis in Mice via the Akt/HIF-1α/VEGF Signaling Pathway"

_molecules, 2023, doi:10.3390/molecules28217238_

Round 1
Reviewer 1 Report
The manuscript describes the pharmacological effects of saffron extract in preventing and treating liver fibrosis through network pharmacology analysis combined with in vivo validation experiments. The work is interesting and contributes to the knowledge in the area. The methods are well-described.
I suggest some minor changes before being accepted for publication.
1. The order of the sections must be changed according to instructions: Introduction, Results, Discussion, Materials and Methods, Conclusions
2. The References are not prepared according to the instructions and should be corrected.
Author Response
The manuscript describes the pharmacological effects of saffron extract in preventing and treating liver fibrosis through network pharmacology analysis combined with in vivo validation experiments. The work is interesting and contributes to the knowledge in the area. The methods are well-described.
I suggest some minor changes before being accepted for publication.
- The order of the sections must be changed according to instructions: Introduction, Results, Discussion, Materials and Methods, Conclusions
Response: Thank you for your suggestion. During the revision process, we have adjusted the order of the sections in the article
- The References are not prepared according to the instructions and should be corrected.
Response: Thank you for your suggestion. We have made modifications according to the required reference format
Reviewer 2 Report
General comments
The paper details a methodological approach in elucidating the mechanism of action of Saffron in treatment of liver fibrosis through network pharmacology and in vivo experiments. Generally the paper is well structured and the methodology quite detailed and thus making this paper a significant scientific contribution. The only drawback I can pick up from the paper is that the title doesn't do justice to the scientific work. I would encourage the authors to amend the title such that it aligns with the aim of the study. Some parts of the methodology are written in present tense and should be written in past tense.
Detailed comments
Line 2. Title - What is the nutritional value being referred to in the title? Amend the title such that the significant finding of the study "modulation of the AKT/HIF-1 alpha/VEGF pathway by Saffron" is also expressed in the title.
Line 17-18. It is not enough to say the 'AKT/HIF-1 alpha/VEGF pathway is significantly affected', provide more details on how the pathway is being affected.
Line 26-30. Include a reference
Line 80-88. This paragraph should be moved to the Materials and Method section.
Line 94. The preparation of Saffron extract should be presented in a past tense indicating how the preparation was performed.
Line 142. Replace calculate with 'calculation of'
The methodology section should be written in past tense since the authors are describing what has already been done.
Author Response
The paper details a methodological approach in elucidating the mechanism of action of Saffron in treatment of liver fibrosis through network pharmacology and in vivo experiments. Generally the paper is well structured and the methodology quite detailed and thus making this paper a significant scientific contribution. The only drawback I can pick up from the paper is that the title doesn't do justice to the scientific work. I would encourage the authors to amend the title such that it aligns with the aim of the study. Some parts of the methodology are written in present tense and should be written in past tense.
Response: Thank you for your suggestion. We have revised the title of the article to "Hepatoprotective Effect of Medicine Food Homology flower Saffron against CCl4 Induced Liver Fibrosis in Mice via the Akt/HIF-1 α/ VEGF Signaling Pathway. We also use the past tense to express the methods in the article.
Detailed comments
Line 2. Title - What is the nutritional value being referred to in the title? Amend the title such that the significant finding of the study "modulation of the AKT/HIF-1 alpha/VEGF pathway by Saffron" is also expressed in the title.
Response: Thank you for your suggestion. We have made modifications to the title of the article.
Line 17-18. It is not enough to say the 'AKT/HIF-1 alpha/VEGF pathway is significantly affected', provide more details on how the pathway is being affected.
Response: Thank you for your suggestion. We will provide detailed explanations during the revision process. “RT-PCR and Western blotting analysis confirmed that saffron treatment prevent CCl4-induced the upregulation of HIF-1 α, VEGFA, AKT, and PI3K suggesting that saffron may regulate AKT/HIF-1 α/ VEGF alleviates liver fibrosis.”
Line 26-30. Include a reference
Response: Thank you for your suggestion. We have added references during the revision process.
Line 80-88. This paragraph should be moved to the Materials and Method section.
Line 94. The preparation of Saffron extract should be presented in a past tense indicating how the preparation was performed.
Line 142. Replace calculate with 'calculation of'
Response: Thank you for your suggestion. We have made modifications during the revision process.
Reviewer 3 Report
In the manuscript entitled "Bioactive compounds, nutritional value and hepatoprotective 2 activity of medicine food homology flower Saffron" authored by Jiang et al, the hepatoprotective properties of Saffron are explored and mechanistically linked to suppression of AKT/Pi3K and subsequent VEG/HIF signaling in fibrosis. The data on construction of networks linking fibrosis to saffron are well-presented.
However, few concerns about the robustness of the western analysis are noted:
1. Even though the authors claim that p-AKT is suppressed by saffron, the repeat experiments shown as raw data are internally inconsistent with each other.
2. The raw data for AKT is cut off and needs to be replaced.
3. The western blot looks like total AKT is also low. What is the ratio of p-AKT/total AKT after saffron treatment? AKT activation is a key player for saffron's effect, but the p-AKT does not clearly represent a significant change. Densitometries for blots should be provided to quantify the changes.
4. Changes in VEGF are not convincing and appear variable in different experiments. Again blot quantification would be required.
Author Response
In the manuscript entitled "Bioactive compounds, nutritional value and hepatoprotective activity of medicine food homology flower Saffron" authored by Jiang et al, the hepatoprotective properties of Saffron are explored and mechanistically linked to suppression of AKT/Pi3K and subsequent VEG/HIF signaling in fibrosis. The data on construction of networks linking fibrosis to saffron are well-presented.
However, few concerns about the robustness of the western analysis are noted:
- Even though the authors claim that p-AKT is suppressed by saffron, the repeat experiments shown as raw data are internally inconsistent with each other.
- The raw data for AKT is cut off and needs to be replaced.
- The western blot looks like total AKT is also low. What is the ratio of p-AKT/total AKT after saffron treatment? AKT activation is a key player for saffron's effect, but the p-AKT does not clearly represent a significant change. Densitometries for blots should be provided to quantify the changes.
- Changes in VEGF are not convincing and appear variable in different experiments. Again blot quantification would be required.
Response: Thank you for your suggestion. During the revision process, we carefully sorted out the Western blot results and supplemented them with experiments. Use Image Pro Plus to calculate the densitometries for blots, and the detailed results are shown in Figure 11 of the article and supplementary documents.
Round 2
Reviewer 3 Report
None